# Long-Term Pulmonary Dysfunction by Hyperoxia Exposure during Severe Viral Lower Respiratory Tract Infection in Mice

**DOI:** 10.3390/pathogens11111334

**Published:** 2022-11-12

**Authors:** Thijs A. Lilien, Miša Gunjak, Despoina Myti, Francisco Casado, Job B. M. van Woensel, Rory E. Morty, Reinout A. Bem

**Affiliations:** 1Pediatric Intensive Care Unit, Emma Children’s Hospital, Amsterdam UMC Location University of Amsterdam, 1105 AZ Amsterdam, The Netherlands; 2Department of Lung Development and Remodeling, Max Planck Institute for Heart and Lung Research, 61231 Bad Nauheim, Germany; 3Department of Internal Medicine (Pulmonology), University of Giessen and Marburg Lung Center (UGMLC), German Center for Lung Research (DZL), 35392 Giessen, Germany; 4Department of Translational Pulmonology, and Translational Lung Research Center (TLRC), German Center for Lung Research (DZL), 69120 Heidelberg, Germany

**Keywords:** hyperoxia, oxygen, bronchiolitis, respiratory tract infection, pneumovirus, respiratory syncytial viruses, airway resistance

## Abstract

Viral-induced lower respiratory tract infection (LRTI), mainly by respiratory syncytial virus (RSV), causes a major health burden among young children and has been associated with long-term respiratory dysfunction. Children with severe viral LRTI are frequently treated with oxygen therapy, hypothetically posing an additional risk factor for pulmonary sequelae. The main goal of this study was to determine the effect of concurrent hyperoxia exposure during the acute phase of viral LRTI on long-term pulmonary outcome. As an experimental model for severe RSV LRTI in infants, C57Bl/6J mice received an intranasal inoculation with the pneumonia virus of mice J3666 strain at post-natal day 7, and were subsequently exposed to hyperoxia (85% O_2_) or normoxia (21% O_2_) from post-natal day 10 to 17 during the acute phase of disease. Long-term outcomes, including lung function and structural development, were assessed 3 weeks post-inoculation at post-natal day 28. Compared to normoxic conditions, hyperoxia exposure in PVM-inoculated mice induced a transient growth arrest without subsequent catchup growth, as well as a long-term increase in airway resistance. This hyperoxia-induced pulmonary dysfunction was not associated with developmental changes to the airway or lung structure. These findings suggest that hyperoxia exposure during viral LRTI at young age may aggravate subsequent long-term pulmonary sequelae. Further research is needed to investigate the specific mechanisms underlying this alteration to pulmonary function.

## 1. Introduction

Viral-induced lower respiratory tract infection (LRTI), most notably by the respiratory syncytial virus (RSV), causes a major global health burden among children below the age of 2 years and as such constitutes one of the most early and important injuries to a young child’s pulmonary system [1,2]. Exposure to viral LRTI at young age is associated with impaired lung function trajectories and long-term sequelae, such as recurrent wheezing and asthma [2,3,4]. Children with severe viral LRTI frequently require respiratory supportive care, necessitating admission to a pediatric intensive care unit (PICU). A recent PICU follow-up study observed previously undiagnosed long-term obstructive respiratory pathology, including asthma, in up to 25% of the children at 6–12 years after severe viral LRTI [5].

A possible contributing risk factor for the development of pulmonary sequelae in young children suffering from severe viral LRTI may be exposure to high-dose oxygen. Oxygen therapy is integral to the treatment of children receiving respiratory support for severe viral LRTI in the PICU [6]. While this supplemental oxygen is undoubtedly a life-saving therapy, its overzealous use has been associated with both acute and long-term detrimental effects, including in critically ill children [7,8,9]. Although long-term effects of hyperoxia exposure in premature infants, such as the development of bronchopulmonary dysplasia, are well established [10], currently little is known about the potential long-term harm of hyperoxia beyond the direct postnatal period. Especially important is the interaction of concurrent hyperoxia exposure with existing lung injury and its potential concomitant adverse effects. Therefore, more insight is needed into the effects of hyperoxia exposure during viral LRTI in young children.

In this study, we aimed to assess the long-term effects of hyperoxia exposure during severe viral LRTI in early life. We hypothesized that hyperoxia exposure during viral LRTI would aggravate long-term pulmonary dysfunction. This was tested in a well-established animal model for severe pediatric viral LRTI using pneumonia virus of mice (PVM) infection in young mice [11].

## 2. Materials and Methods

### 2.1. Animals and Virus Strain

Timed-pregnant C57Bl/6J mice were obtained from Janvier Labs (Laval, France) and were housed in a temperature-controlled room on a 12 h/12 h light-dark cycle. Dams and pups received food and water ad libitum. The pathogenic PVM strain J3666, originally obtained from the Rockefeller University, was continuously passaged in mice to maintain virulence [12]. Viral stock aliquots contained 1.86 × 10^9^ copies of PVM per mL.

### 2.2. Animal Protocols

Newborn pups (born within 12 h of each other) were randomized and divided in equal numbers over all dams. Sex of the newborn mice was determined by investigation of the external genitalia [13]. At post-natal day (P) 7 the pups received an intranasal inoculation of either PVM diluted in RPMI 1640 medium (Thermo Fisher Scientific, Waltham, MA, USA) or RPMI 1640 medium only (non-infected control) in a total volume of 10 µL. The optimal PVM dose for the long-term model was determined through pilot experiments. Ultimately, a dose of 1:750 was used for PVM inoculation to ensure considerable morbidity during the acute phase, but also survival for assessment of long-term outcomes. Consequently, this dilution and inoculum volume resulted in an intranasal inoculum of 2.5 × 10^4^ copies of PVM. Subsequently, both PVM-infected pups and non-infected controls were kept in room air until P10, after which the litters were randomized 1:1 to be maintained in hyperoxia (85% O_2_, [HYX]) in a specially designed chamber or in room air/normoxia (21% O_2_, [NOX]). As such, four groups of mice were established: NOX-RPMI (N = 30), HYX-RPMI (N = 28), NOX-PVM (N = 28), HYX-PVM (N = 31). The hyperoxia-exposed litters were kept in their hyperoxic environment until P17, after which the litters returned to room air. During hyperoxia exposure, dams were exchanged daily with room air litters to reduce oxygen toxicity in the dams. Follow-up occurred through P28, at which the mice were euthanized by an intraperitoneal injection of 500 mg.kg^−1^ pentobarbital (Narcoren; Boehringer Ingelheim, Ingelheim am Rhein, Germany) for end-point outcome measurements. During pilot experiments for dose finding, mice were euthanized at P14 to assess viral replication 7 days post-infection. Total body weight of the pups was measured daily.

### 2.3. Measurements Viral Load

Lungs were isolated at P14, the left sided single lobe was homogenized in lysis buffer (peqGold Total RNA Kit; VWR, Radnor, PA, USA) and total RNA was isolated according to manufacturer’s instructions. cDNA synthesis was performed with 1000 ng of total RNA using M-MLV reverse transcriptase (Thermo Fisher Scientific, Waltham, MA, USA) and random hexamers (Thermo Fisher Scientific, Waltham, MA, USA). Copies of the PVM-small hydrophobic (*Sh*) gene were detected in quantitative real-time PCR reactions containing 2 µL of cDNA, SYBR Green qPCR Master Mix (Thermo Fisher Scientific, Waltham, MA, USA) and primers (5′-CCGTCATCAACACAGTGTGT-3′ and 5′-GCCTGATGTGGCAGTGCTT-3′; Eurofins Genomics, Luxemburg) [14]. The *POLR2A* housekeeping gene was detected in cDNA samples using the primers (5′-CTAAGGGGCAGCCAAAGAAAC-3′ and 5′-CCATTCAGCATACAACTCTAGGC-3′; Eurofins Genomics, Luxemburg). An additional standard curve with known concentrations of the full-length *Sh* gene was used for quantification of the viral load of the viral stock, as described before [15].

### 2.4. Pulmonary Function

Pulmonary function parameters were analyzed with the use of a FlexiVent system (SCIREQ, Montréal, Canada). Mice were anesthetized with an intraperitoneal injection of 80 mg.kg^−1^ ketamine (Medistar, Houston, TX, USA) and 12 mg.kg^−1^ xylazine (Bayer HealthCare, Leverkusen, Germany). Tracheal cannulation was performed using a 20-gauge blunt needle (CML Supply, Lexington, KY, USA). Subsequently, an intraperitoneal injection of 1 mg.kg^−1^ pancuronium (Inresa Arzneimittel, Freiburg, Germany) was administered and positive pressure ventilation was initiated. Mice were briefly ventilated at 150 breaths per minute for the duration of measurements, with 10 mL.kg^−1^ tidal volume and 3 cmH_2_O positive end-expiratory pressure, as these settings cause relatively little short-term ventilator-induced lung injury [15,16]. Before measurement of respiratory parameters the lungs were recruited by inflating the lungs to their total inspiratory capacity. Subsequently, inspiratory capacity; single frequency forced oscillation technique (FOT) (SnapShot-150), to assess the total respiratory system; input impedance and broadband FOT (Quick Prime-3), to distinguish between airway and tissue mechanics; and pressure-driven pressure-volume loops were acquired five times in a repetitive cycle and values were averaged. Technically correctly obtained measurements with a coefficient of determination > 0.95 were considered as valid. Invalid measurements were omitted from the analysis.

### 2.5. Alveolar Structural Analysis and Quantification

For the alveolar structural analysis we adhered to the American Thoracic Society/European Respiratory Society guidelines for quantitative assessment of lung structure and main principles of design-based stereology [17,18,19]. The isolation and fixation of lungs for design-based stereology was performed as described before [20]. In short, lungs of mice were inflated at a hydrostatic pressure of 20 cmH_2_O with a 1.5% paraformaldehyde (PFA) (Sigma-Aldrich, Darmstadt, Germany), 1.5% glutaraldehyde (Serva Electroforesis, Heidelberg, Germany), 0.15 M HEPES (Sigma-Aldrich, Darmstadt, Germany) in phosphate-buffered saline (PBS) solution (pH 7.4) (Sigma-Aldrich, Germany). Subsequently, lungs were embedded in 2% agar (Sigma-Aldrich, Darmstadt, Germany) and cut into 3 mm sections to determine the total lung volume by the Cavalieri principle [21]. All lung parts were then treated with 0.1 M sodium cacodylate trihydrate (Serva Electroforesis, Heidelberg, Germany), 1% osmiumtetroxide (Carl Roth, Karlsruhe, Germany), 2.5% uranyl acetate (Serva Electroforesis, Heidelberg, Germany), acetone dilutions and finally embedded in Technovit 7100 (Kulzer, Hanau, Germany). Sections of 2 µm were acquired, four sections of every tenth consecutive section were used to analyze all parameters besides alveoli number, for the latter a first and third section were used. Sections were stained with Richardson’s stain and afterwards visualized using a digital slide scanner (NanoZoomer-XR C12000; Hamamatsu Photonics, Shizuoka, Japan) [22]. Quantitative analysis was performed using Visiopharm’s newCAST program (Visiopharm A/S, Hørsholm, Denmark). Mean linear intercept, alveolar septal thickness, surface area of gas exchange, alveolar density and alveolar number were analyzed.

### 2.6. Lung Histology

After necropsy, the lungs were isolated and inflated with 4% PFA at a hydrostatic pressure of 20 cmH_2_O (Sigma-Aldrich, Darmstadt, Germany) in PBS (Sigma-Aldrich, Darmstadt, Germany) and stored overnight in 4% PFA at 4 °C. Lungs were then cleaned and embedded in paraffin blocks. Two sections of 5 µm were cut from paraffin-embedded lungs of five mice per group and stained with Mayer’s hematoxylin (Sigma-Aldrich, Darmstadt, Germany) and eosin (Sigma-Aldrich, Darmstadt, Germany) (H&E), and with Masson’s Trichrome (MT) staining (Morphisto, Offenbach am Main, Germany) to visualize collagen. To analyze histological parameters, ten round to (semi-) oval airways were randomly selected per mouse and visualized (Olympus UC90; Olympus Corporation, Tokyo, Japan). Subsequently, the visualized airways were blinded to the observer before quantitative analysis using a random sequence number to minimize observer bias. Epithelial thickness was determined by averaging the epithelial thickness measured in each quadrant of the airway. Airway base perimeter and inner airway circumference were also measured. To quantify collagen staining specifically in the airways, the total area of collagen was determined with the use of digital image software (ImageJ; NIH, Bethesda, MD, USA), using a color threshold of 125–225. The total area of collagen was normalized to the airway base perimeter (collagen area/base perimeter) and the total airway area (collagen area/airway area). The acquired values of airways were averaged to obtain a single value per mouse.

### 2.7. Immunohistochemistry

Two paraffin-embedded lung sections of 5 µm were acquired from five animals per group. Sections were deparaffinized and rehydrated. Immunohistochemistry was performed by blocking all sections in a 3% H_2_O_2_ solution (Sigma-Aldrich, Darmstadt, Germany) for 15 min to block endogenous peroxidase activity. Antigen retrieval was performed using a citric acid buffer solution (Sigma-Aldrich, Darmstadt, Germany) for 45 min. Sections were blocked afterwards with 10% goat serum (Biowest, Nuaillé, France) in 1% bovine serum albumin (BSA) (Sigma-Aldrich, Darmstadt, Germany) tris-buffered saline (TBS) solution for 40 min. Sections were then incubated overnight at 4 °C with the primary monoclonal rabbit antibody to α-smooth muscle actin (α-SMA) (Sigma-Aldrich, Darmstadt, Germany) diluted 1:100 in 1% BSA TBS solution or with 1% BSA TBS solution only, as a negative control. Subsequently, sections were incubated with the horseradish peroxidase-conjugated (HRP) secondary polyclonal goat-anti-rabbit antibody (Abcam, Cambridge, UK) for 1 h using a 1:2000 dilution and sections were afterwards developed for 5 min with 3,3′-diaminobenzidine tetrahydrochloride (DAB) substrate (Sigma-Aldrich, Darmstadt, Germany) as the chromogen. Finally, sections were counterstained with hematoxylin, dehydrated and mounted. Ten airways were randomly selected per mouse, visualized and blinded to the observer as described above. Airway smooth muscle (ASM) area was analyzed by quantification of the α-SMA-positively stained area using digital image software (ImageJ; NIH, Bethesda, MD, USA) and was normalized to airway base perimeter and total airway area.

### 2.8. Statistical Analysis

Results are reported as median with interquartile range (IQR) or as mean with standard deviation (SD) or 95% confidence interval (CI). The Log-rank test was used to assess equality in survival between conditions. One-way ANOVA with Tukey’s post hoc multiple comparison correction was used to test for differences between groups. To estimate the precision of the design-based stereology data, the coefficient of error (CE), the coefficient of variation (CV) and the squared ratio between those (CE^2^/CV^2^) was measured for each parameter. The quotient threshold (CE^2^/CV^2^) was set at < 0.5 to validate the precision of the measurements. A *p* value of < 0.05 was considered as statistically significant. All statistical analyses were performed using Prism 9 (GraphPad Software, San Diego, CA, USA).

## 3. Results

### 3.1. Establishing the Model

In the pilot experiments, replication of the PVM-Sh gene was observed at 7 days post-intranasal inoculation for multiple doses (Appendix A).

### 3.2. Exposure to Hyperoxia during PVM Infection Leads to Growth Restriction

In follow-up studies, survival did not differ between experimental groups at the end-point (P28) with the selected viral dose (Appendix A). PVM-infected mice exposed to hyperoxia exhibited a transient growth arrest without catch-up growth after clinical resolution of infection (Figure 1). In more detail, the growth curve of this group started to deviate from the control group 7 days after inoculation (P14) and this growth arrest continued up to the 13th day post-inoculation (P20), which was 3 days after completion of hyperoxia exposure. At the latter time-point the difference in total body weight was the most pronounced compared with all other groups (mean weight difference HYX-PVM vs. NOX-RPMI: −1.3 g, 95% CI −0.8–−1.9; HYX-RPMI: −0.9 g, 95% CI −0.4–−1.5; NOX-PVM −1.1 g, 95% CI −0.6–−1.7; *p* value < 0.001). By the end-point (P28) mean total body weight of the HYX-PVM mice was still significantly lower compared to the other conditions (mean weight difference HYX-PVM vs. NOX-RPMI: −1.5 g, 95% CI −0.5–−2.4; HYX-RPMI: −1.4 g, 95% CI −0.5–−2.4; NOX-PVM −1.1 g, 95% CI −0.1–−2.0; *p* value < 0.001).

### 3.3. Hyperoxia Exposure during PVM Infection Alters Long-Term Pulmonary Function

When pulmonary function was analyzed at P28, 2 invalid measurements were omitted from each of the HYX-RPMI and NOX-PVM groups based on technical validity defined by the coefficient of determination (included in analysis: NOX-RPMI: N = 9; HYX-RPMI: N = 6; NOX-PVM: N = 6; HYX-RPMI: N = 9). Deep inflation revealed no difference in inspiratory capacity between conditions (Appendix A). Single frequency FOT revealed no difference in the dynamic compliance (Crs) or elastance of the respiratory system (Ers) (Figure 2). However, the resistance of the respiratory system (Rrs) was significantly increased in HYX-PVM mice compared to both non-infected conditions (mean Rrs difference HYX-PVM vs. NOX-RPMI: 0.26 cmH_2_O.s/mL, 95% CI 0.04–0.48, *p* value = 0.018; HYX-RPMI: 0.30 cmH_2_O.s/mL, 95% CI 0.05–0.54, *p* value = 0.015; NOX-PVM: 0.18 cmH_2_O.s/mL, 95% CI −0.06–0.43, *p* value = 0.199) (Figure 2). Moreover, broadband frequency FOT revealed a substantial and significant increase in the airway resistance (R_N_) in HYX-PVM mice, which was higher compared to all other conditions (mean R_N_ difference HYX-PVM vs. NOX-RPMI: 0.19 cmH_2_O.s/mL, 95% CI 0.06–0.32, *p* value = 0.002; HYX-RPMI: 0.24 cmH_2_O.s/mL, 95% CI 0.10–0.39, *p* value < 0.001; NOX-PVM 0.19 cmH_2_O.s/mL, 95% CI 0.05–0.34, *p* value = 0.005) (Figure 3). Tissue damping (G) and tissue elastance (H) did not differ between groups (Figure 3). Additionally, visual assessment of the impedance, specifically the real impedance (Zrs), showed that the moderate increase in impedance in the HYX-PVM group was relatively consistent over low and high frequencies (Figure 4). Assessment of the pressure-volume loops did not reveal meaningful differences between the conditions (Appendix A).

### 3.4. Alveolar Development Is Not Impaired by Hyperoxia Exposure during PVM Infection

In the structural analysis of the lungs, no difference in lung volume was observed between the groups. As indicated in Table 1, assessment of the alveolar morphology and lung structure did not reveal any differences in the septal thickness, mean linear intercept or gas exchange surface area (Figure 5). Quantification of the number of alveoli indicated a small decrease in alveolar density in both hyperoxia-exposed groups compared to the NOX-PVM group, however, no difference was observed with the NOX-RPMI group. Yet, the total alveoli number did not differ between groups (Table 1).

### 3.5. Hyperoxia Exposure during PVM Infection Does Not Lead to Airway Remodeling

In the histologic analysis of the airways, no sign of ongoing inflammation at day 28 was observed in any of the experimental groups (Figure 6). Captured airways varied in size and ranged from 268 to 805 µm in base perimeter (median 462 µm, IQR 392–529). As indicated in Table 2, epithelial thickness of the airways did not differ between experimental groups. Similarly, no change was observed in the inner airway circumference (inner airway circumference/outer airway circumference). Additionally, quantitative analysis of the airways revealed no difference between groups in the total area of collagen or airway smooth muscle area, either normalized to the airway perimeter or airway area (Figure 7, Table 2). These observations were maintained in a sensitivity analysis including smaller airways only (<500 µm base perimeter; data not shown).

## 4. Discussion

The main goal of this study was to assess the long-term effects of hyperoxia exposure during viral LRTI in early life. In this study, hyperoxia exposure during viral LRTI induced a transient growth arrest without catchup growth after resolution of infection. Furthermore, exposure to hyperoxia resulted in long-term pulmonary dysfunction in PVM-inoculated mice, namely increased airway resistance.

These findings contribute to our knowledge on the long-term effects of high dose oxygen treatment during viral LRTI at young age. As an experimental model, a well-established PVM model in young mice was employed to mimic severe human viral LRTI by RSV in infants [11]. This cognate host-pneumovirus model mirrors many of the specific features of RSV-LRTI including clinical disease, histopathology and pro-inflammatory profiles. The mice were inoculated with PVM at P7, which is approximately the human equivalent of a full-term neonate at birth up to the first months of age regarding the developmental stage of the lungs [20,23]. As such, this resembled infection in early life. Subsequently, long-term outcomes were assessed at P28, which is considered early adolescence in mice and approximately reflects the same stage of life in humans [2,20,24]. Previously, viral LRTI in early life has been associated with long-term pulmonary dysfunction in humans, most notably recurrent wheezing and asthma in later life [3,5]. However, whether this long-term pulmonary dysfunction has a causal relationship with early life viral infection is still a matter of debate [4]. Here, it is proposed that treatment with high dose oxygen might pose an additional risk factor to the development of long-term pulmonary dysfunction in subjects exposed to viral LRTI in early life.

Other reports that have employed mouse models to study the long-term effects of viral LRTI have considered viral infection primarily as an isolated risk factor [25,26,27]. These studies have reported mixed results: some documented pulmonary dysfunction only in the acute phase of disease [25,26], whereas another PVM-based model observed long-term increased airway hyperresponsiveness but only upon methacholine provocation [27]. In the present study, PVM inoculation with subsequent hyperoxia exposure affected baseline lung function, whereas room air-exposed PVM-inoculated mice did not reveal an altered baseline airway resistance, without specifically provoking airway hyperresponsiveness. This suggests that concurrent hyperoxia exposure may aggravate the previously observed effects of viral LRTI on airway resistance. In this same line, concurrent hyperoxia exposure in animal models of ventilator-induced lung injury also has potential synergistic effects on outcome, exacerbating the observed pulmonary injury [28,29,30]. In addition, in neonatal viral LRTI models with rhinovirus inoculation following prolonged hyperoxia exposure of immature P2–3 mice, exposure to hyperoxia preceding infection aggravated viral-induced lung inflammation and airway hyperresponsiveness in the acute phase [31]. However, to the authors’ knowledge, no comparative models exist examining hyperoxia exposure during the acute phase of viral LRTI at young age.

In light of the previously described association of recurrent wheezing and asthma after viral LRTI in early life [3,5], it is interesting to note that the observed pulmonary dysfunction in our study resembles previous findings of pulmonary dysfunction in asthmatic subjects [32,33,34,35]. More specifically, a moderate increase in impedance over both low and high frequencies, thus an increased airway resistance, without evident changes in lung elastance have been observed in mild-to-moderate asthma [32,33,34]. Additionally, an increase in baseline airway resistance has also been described in subjects with mild-to-moderate asthma [35]. In our study, hyperoxia-treated mice demonstrated persistent growth problems (no catchup of growth arrest) after the acute phase of viral LRTI, suggesting that the hyperoxia-induced impairment in pulmonary function was clinically relevant. Although we observed pulmonary dysfunction similar to that of patients with mild-to-moderate asthma, we could not observe airway remodeling that could potentially explain this increase in airway resistance. However, airway remodeling does not necessarily translate to severity of pulmonary dysfunction [36]. Furthermore, any possible airway remodeling present might have been too heterogeneously distributed to capture this easily from a set of randomly selected airways. Lastly, the increase in airway resistance may also have resulted from alterations in the immunomodulatory response [31]. Therefore, future studies should focus on the potential mechanisms underlying this increase in airway resistance, such as the immunomodulatory response or by visualizing the total respiratory system using 3D computed tomography to assess heterogeneously distributed airway narrowing [37].

## 5. Limitations

Limitations of this study need to be acknowledged. First, although PVM clinically mimics RSV-induced viral LRTI in humans well compared to RSV administration to mice, PVM remains a different pathogen to those found as causative pathogens of viral LRTI in humans. Second, previous mouse models of PVM have predominantly used BALB/c mice as these are more susceptible to PVM [38]. However, BALB/c mice are also known to be very sensitive to the acute effects of oxygen toxicity [39]. Therefore, C57Bl/6J was selected as the strain of choice, which is more commonly used in hyperoxia models, to ensure long-term survival. Third, given the number of labor-intensive measurements, airway hyperresponsiveness to methacholine provocation was not investigated, and only comprehensive baseline lung function parameters were determined. However, as a consequence, increased airway hyperresponsiveness in any of the experimental groups may have been missed.

## 6. Conclusions

In this study using a well-establish murine model of viral LRTI, concurrent hyperoxia exposure during the acute phase of viral LRTI in infant mice was associated with growth restriction and increased airway resistance after resolution of the infection, while structural pulmonary development appeared unaffected. Based on these findings, it is speculated that exposure to hyperoxia might be a contributing factor to the previously observed association of severe viral LRTI in young children and long-term pulmonary sequelae. Further research is needed to clarify the underlying mechanisms of this abnormal lung development.

## Figures and Tables

**Figure 1 pathogens-11-01334-f001:**
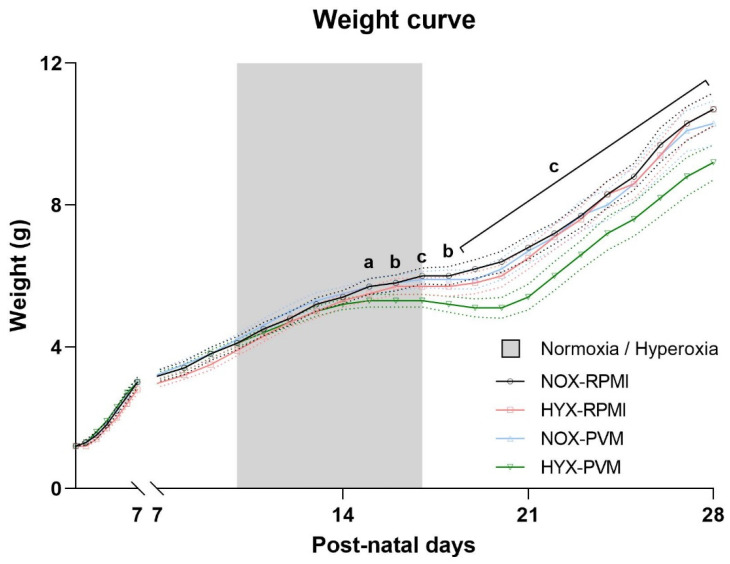
Daily total body weight of mice per group: RPMI non-infected controls and PVM-infected mice with either normoxia exposure (NOX: 21% O_2_) or hyperoxia exposure (HYX: 85% O_2_). Grey area represents the duration of hyperoxia exposure. Mean weight with 95% CI are shown. NOX-RPMI (N = 30), HYX-RPMI (N = 28), NOX-PVM (N = 28), HYX-PVM (N = 31). a, *p* < 0.05 HYX-PVM vs. NOX-RPMI; b, *p* < 0.05 HYX-PVM vs. NOX-RPMI, HYX-RPMI; c, *p* < 0.05 HYX-PVM vs. all conditions.

**Figure 2 pathogens-11-01334-f002:**
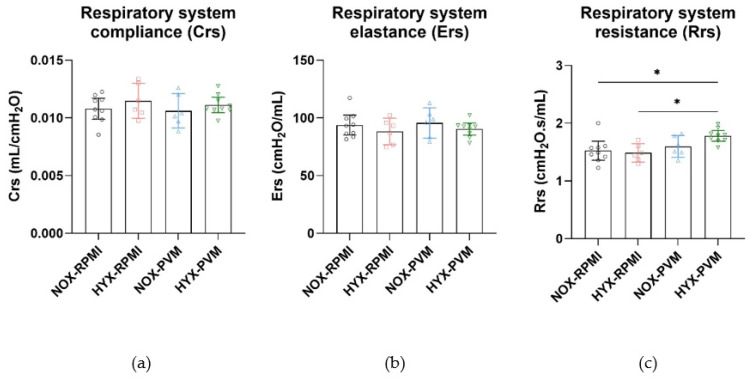
Parameters based on single frequency FOT: (**a**) Compliance (Crs); (**b**) Elastance (Ers); and (**c**) Resistance (Rrs) of the respiratory system per group. RPMI non-infected controls and PVM-infected mice with either normoxia exposure (NOX: 21% O_2_) or hyperoxia exposure (HYX: 85% O_2_). Mean with 95% CI are shown. NOX-RPMI (N = 9), HYX-RPMI (N = 6), NOX-PVM (N = 6), HYX-PVM (N = 9). *, *p* < 0.05.

**Figure 3 pathogens-11-01334-f003:**
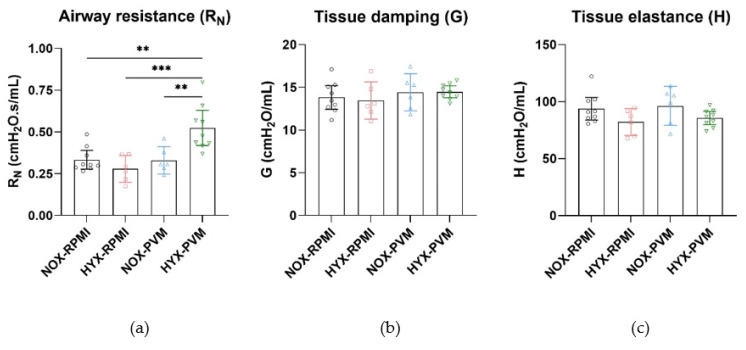
Parameters based on broadband frequency FOT: (**a**) Airway resistance (Rn); (**b**) Tissue damping (G); and (**c**) Tissue elastance (H) per group. RPMI non-infected controls and PVM-infected mice with either normoxia exposure (NOX: 21% O_2_) or hyperoxia exposure (HYX: 85% O_2_). Mean with 95% CI are shown. NOX-RPMI (N = 9), HYX-RPMI (N = 6), NOX-PVM (N = 6), HYX-PVM (N = 9). **, *p* < 0.01; ***, *p* < 0.001.

**Figure 4 pathogens-11-01334-f004:**
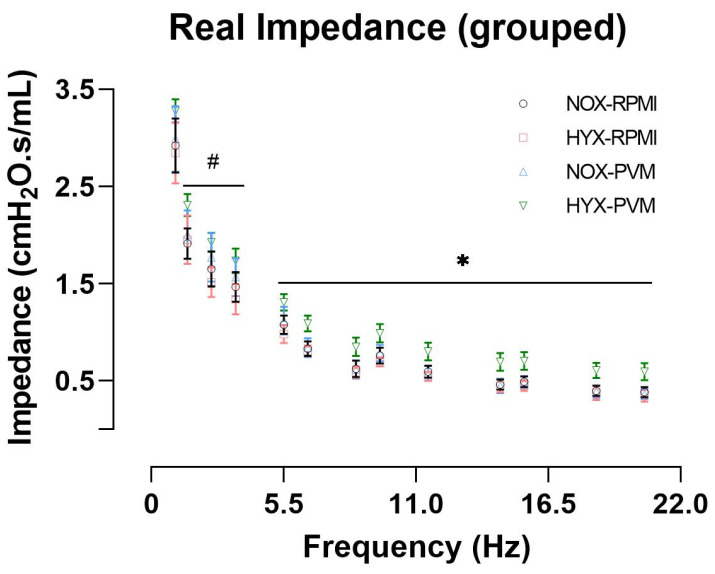
Real impedance based on broadband frequency FOT per group: RPMI non-infected controls and PVM-infected mice with either normoxia exposure (NOX: 21% O_2_) or hyperoxia exposure (HYX: 85% O_2_). Mean with 95% CI are shown. NOX-RPMI (N = 9), HYX-RPMI (N = 6), NOX-PVM (N = 6), HYX-PVM (N = 9). # *p* < 0.05 HYX-PVM vs. NOX-RPMI and HYX-RPMI, * *p* < 0.05 HYX-PVM vs. all conditions.

**Figure 5 pathogens-11-01334-f005:**
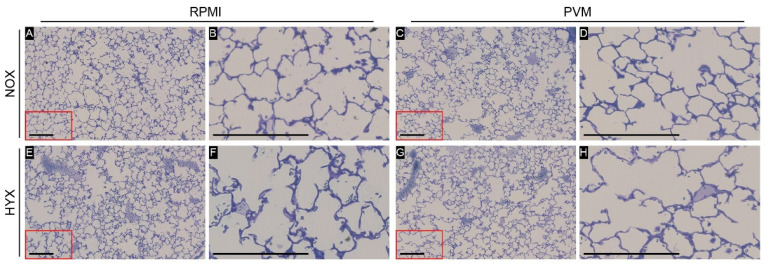
Representative pictures of design-based stereology per group: RPMI non-infected controls and PVM-infected mice with either normoxia exposure (NOX: 21% O_2_) or hyperoxia exposure (HYX: 85% O_2_). N = 5 for all conditions. (**A**,**C**,**E**,**G**): 100× magnification; (**B**,**D**,**F**,**H**): 400× magnification. Scale bar represents 200 µm.

**Figure 6 pathogens-11-01334-f006:**
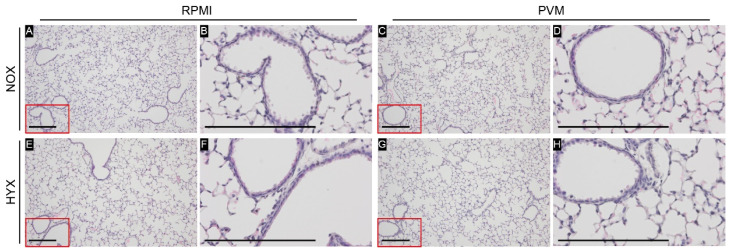
Representative pictures of H&E staining per group: RPMI non-infected controls and PVM-infected mice with either normoxia exposure (NOX: 21% O_2_) or hyperoxia exposure (HYX: 85% O_2_). N = 5 for all conditions. (**A**,**C**,**E**,**G**): 100× magnification; (**B**,**D**,**F**,**H**): 400× magnification. Scale bar represents 200 µm.

**Figure 7 pathogens-11-01334-f007:**
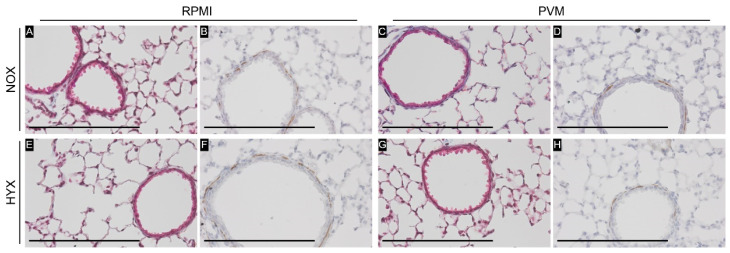
Representative pictures of (**A**,**C**,**E**,**G**): MT staining for collagen (blue) and (**B**,**D**,**F**,**H**): immunohistochemistry staining for α-SMA (brown) per group: RPMI non-infected controls and PVM-infected mice with either normoxia exposure (NOX: 21% O_2_) or hyperoxia exposure (HYX: 85% O_2_). N = 5 for all conditions. All pictures are at 400 × magnification. Scale bar represents 200 µm.

**Table 1 pathogens-11-01334-t001:** Stereological parameters determined for the different conditions of viral infection and oxygen exposure at post-natal day 28.

			NOX-RPMI			HYX-RPMI			NOX-PVM			HYX-PVM			ANOVA *p* Value
			Mean ± SD			Mean ± SD			Mean ± SD			Mean ± SD			
V (lung) [cm^3^]			0.376 ± 0.036			0.368 ± 0.023			0.337 ± 0.031			0.349 ± 0.015			0.1432
CE	CV	CE^2^/CV^2^	0.043	0.096	0.200	0.028	0.062	0.200	0.041	0.091	0.200	0.019	0.043	0.200	
Vv (par/lung) [%]			87.310 ± 1.301			86.930 ± 1.159			86.870 ± 0.871			87.870 ± 0.861			0.4457
CE	CV	CE2/CV2	0.007	0.015	0.200	0.006	0.013	0.200	0.004	0.010	0.200	0.004	0.010	0.200	
N (alv, lung) ×10^6^			5.830 ± 0.764			5.433 ± 0.333			5.272 ± 0.306			5.234 ± 0.263			0.2011
CE	CV	CE2/CV2	0.059	0.131	0.200	0.027	0.061	0.200	0.026	0.058	0.200	0.022	0.050	0.200	
Nv (alv/par) ×10^6^ [cm^−3^]			17.730 ± 0.681			16.970 ± 0.391 (*p* = 0.0168 vs. NOX-PVM)			18.030 ± 0.561			17.090 ± 0.169 (*p* = 0.0361 vs. NOX-PVM)			0.0095 *
CE	CV	CE2/CV2	0.017	0.038	0.200	0.010	0.023	0.200	0.014	0.031	0.200	0.004	0.010	0.200	
S (alv epi, lung) [cm^2^]			269.900 ± 29.430			264.000 ± 13.540			239.400 ± 18.770			250.900 ± 10.710			0.1010
CE	CV	CE2/CV2	0.049	0.109	0.200	0.023	0.051	0.200	0.035	0.078	0.200	0.019	0.043	0.200	
Sv (alv epi/par) [cm^−1^]			822.100 ± 12.730			824.900 ± 14.710			817.600 ± 17.050			819.400 ± 13.410			0.8672
CE	CV	CE2/CV2	0.007	0.015	0.200	0.008	0.018	0.200	0.009	0.021	0.200	0.007	0.016	0.200	
τ (sep) [µm]			6.948 ± 0.437			7.442 ± 0.312			7.403 ± 0.320			7.363 ± 0.105			0.0859
CE	CV	CE2/CV2	0.028	0.063	0.200	0.019	0.042	0.200	0.019	0.043	0.200	0.006	0.014	0.200	
MLI [µm]			34.770 ± 0.998			33.620 ± 1.226			34.130 ± 1.438			34.100 ± 0.737			0.4764
CE	CV	CE2/CV2	0.013	0.029	0.200	0.016	0.036	0.200	0.019	0.042	0.200	0.010	0.022	0.200	

RPMI non-infected controls and PVM-infected mice with either normoxia exposure (NOX: 21% O_2_) or hyperoxia exposure (HYX: 85% O_2_). Mean ± SD are shown; N = 5 lungs per group. *, Tukey’s post-hoc multiple comparison test was performed after significant one-way ANOVA. Alv, alveoli; Alv epi, alveolar epithelium; CE, coefficient of error; CV coefficient of variance; MLI, mean linear intercept; N, number; Nv, numerical density; Par, parenchyma; S, surface area; Sv, surface density; τ, arithmetic mean septal thickness; V, volume; Vv, volume density.

**Table 2 pathogens-11-01334-t002:** Quantitative parameters of structural airway analysis determined for the different conditions of viral infection and oxygen exposure at post-natal day 28.

	NOX-RPMI	HYX-RPMI	NOX-PVM	HYX-PVM	ANOVA *p* Value
	Mean ± SD	Mean ± SD	Mean ± SD	Mean ± SD	
Structural airway properties:					
Epithelial thickness (µm)	8.160 ± 0.688	8.000 ± 0.860	8.980 ± 1.375	8.340 ± 0.737	0.4130
Normalized to base perimeter (µm.µm^−1^)	0.019 ± 0.002	0.019 ± 0.002	0.020 ± 0.003	0.018 ± 0.003	0.7297
Inner airway / Outer airway (%)	80.3 ± 1.0	80.3 ± 1.7	78.8 ± 1.5	80.4 ± 2.0	0.3745
Area of collagen:					
Normalized to base perimeter (px.µm^−1^)	37 ± 21	81 ± 38	59 ± 33	56 ± 24	0.1943
Normalized to airway area (%)	1.0 ± 0.5	1.7 ± 1.0	2.4 ± 1.2	1.5 ± 0.5	0.1408
αSMA positive area:					
Normalized to base perimeter (px.µm^−1^)	62 ± 13	54 ± 22	86 ± 27	93 ± 30	0.0557
Normalized to airway area (%)	1.7 ± 0.5	1.4 ± 0.6	2.2 ± 0.6	2.2 ± 0.7	0.1336

RPMI non-infected controls and PVM-infected mice with either normoxia exposure (NOX: 21% O_2_) or hyperoxia exposure (HYX: 85% O_2_). Mean ± SD are shown; *N* = 5 lungs per group. αSMA, α-smooth muscle actine.

## Data Availability

Data supporting the findings of this study are available upon reasonable request from the corresponding author, T.A.L.

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
