# Peer review of "Long-Term Pulmonary Dysfunction by Hyperoxia Exposure during Severe Viral Lower Respiratory Tract Infection in Mice"

_pathogens, 2022, doi:10.3390/pathogens11111334_

Round 1
Reviewer 1 Report
This technical and detailed study examines the effects of hyperoxia on total body weight and pulmonary function in mice with an acute viral lower respiratory tract infection. Pulmonary function as measured is permanently altered following hyperoxic exposure. The data is well presented and clearly analysed, and supports the conclusions given and discussed. Comparing a murine model to the observations made in the human clinical setting is very useful and will allow some hypothesis testing as to the aetiology of the effects of acute hyperoxic oxygen exposure in the neonate and young child.
Reviewer 2 Report
The manuscript submitted for review has a good idea as to its purpose.
However, in general, the manuscript has a certain anarchy in its composition.
The number of keywords can also be improved. Do not use phrases but simple words.
They present an introduction where the authors could put the topic a little more in context.
From the introduction, they go directly to RESULTS.
At this point, the authors have to redistribute the manuscript and put the materials and methods in point 2 and not in point 4, which is where it currently is.
About the materials and methods. These are well described.
Results.
In this part, the reading of the same sometimes seems that we are reading part of the material and methods rather than results, even with phrases that are or are part of conclusions.
It is necessary that the results directly expose the results without additions that are not part of this section.
The photo captions are very extensive and the information given in the results is repeated. Sometimes it even seems more like the captions are the results rather than a description of the figure or table.
All the captions of the photo, figure, and table would be modified. They must be a sentence, not a continuum of results.
Within the discussion, they place the limitations of the study.
This part of limitations should go out of the discussion in its own section
Likewise, the last section of the discussion seems to be the conclusions of the work. The manuscript does not indicate conclusions, although it seems that this paragraph may be the end of the discussion.
In a study like this where the conclusions section must be clear and where they are exposed. Although they can be part of the discussion, I believe that the work would improve and even the final understanding of it if the authors explained these clearly. I would give a greater value to the manuscript and not, as they are currently, part of the discussion, a losing entity.
When reconstructing the manuscript, the authors must also modify the numbering of the bibliography.
Round 2
Reviewer 2 Report
The changes that the authors have proposed in this second submission further clarify the manuscript.
The indicated changes are correct.